# Ecological momentary assessment of physical and eating behaviours: The WEALTH feasibility and optimisation study with recommendations for large-scale data collection

**Michael Janek**[1☯], **Jitka Kuhnova**[2☯], **Greet Cardon**[3], **Delfien Van Dyck**[3], **Richard Cimler**[2], **Steriani Elavsky**[4], **Leopold K. Fezeu**[5], **Jean-Michel Oppert**[5,6], **Christoph Buck**[7], **Antje Hebestreit**[7], **Janas Harrington**[8], **Luis Sigcha**[9], **Pepijn Van de Ven**[9], **Alan Donnelly**[9], **Tomas Vetrovsky**[1,2]*, **on behalf of the WEALTH consortium**[¶]

1 Faculty of Physical Education and Sport, Charles University, Prague, Czechia, 2 Faculty of Science, University of Hradec Kralove, Hradec Kralove, Czechia, 3 Department of Movement and Sports Sciences, Ghent University, Ghent, Belgium, 4 Department of Human Movement Studies, University of Ostrava, Ostrava, Czechia, 5 Nutritional Epidemiology Research Team (EREN), Université Sorbonne Paris Nord (USPN), INSERM, INRAE, CNAM, Center of Research in Epidemiology and StatisticS (CRESS), Bobigny, France, 6 Department of Nutrition, Pitié-Salpêtrière Hospital, Assistance Publique-Hôpitaux de Paris (AP-HP), Sorbonne University, CRNH-Ile de France, Paris, France, 7 Leibniz Institute for Prevention Research and Epidemiology—BIPS, Bremen, Germany, 8 School of Public Health, University College Cork, Cork, Ireland, 9 Health Research Institute, University of Limerick, Limerick, Ireland

☯ These authors contributed equally to this work.
¶ The complete membership of the author group can be found in the Acknowledgments
* tomas.vetrovsky@gmail.com

**Data Availability Statement:** All data are available at the Open Science Framework [OSF] repository

## Abstract

Ecological Momentary Assessment (EMA) enables the real-time capture of health-related behaviours, their situational contexts, and associated subjective experiences. This study aimed to evaluate the feasibility of an EMA targeting physical and eating behaviours, optimise its protocol, and provide recommendations for future large-scale EMA data collections. The study involved 52 participants (age 31±9 years, 56% females) from Czechia, France, Germany, and Ireland completing a 9-day free-living EMA protocol using the HealthReact platform connected to a Fitbit tracker. The EMA protocol included time-based (7/day), event-based (up to 10/day), and self-initiated surveys, each containing 8 to 17 items assessing physical and eating behaviours and related contextual factors such as affective states, location, and company. Qualitative insights were gathered from post-EMA feedback interviews. Compliance was low (median 49%), particularly for event-based surveys (median 34%), and declined over time. Many participants were unable or unwilling to complete surveys in certain contexts (e.g., when with family), faced interference with their daily schedules, and encountered occasional technical issues, suggesting the need for thorough initial training, an individualised protocol, and systematic compliance monitoring. The number of event-based surveys was less than desired for the study, with a median of 2.4/day for sedentary events, when 4 were targeted, and 0.9/day for walking events, when 3 were targeted.

(URL: https://osf.io/6dfqk/; DOI 10.17605/OSF.IO/6DFQK).

**Funding:** The WEALTH Project is funded by the Joint Programming Initiative (JPI) Healthy Diet for a Healthy Life (HDHL), a research and innovation initiative of EU member states and associated countries, under grant agreement No 727565, under STAMIFY (Standardised measurement, monitoring and/or biomarkers to study food intake, physical activity and health). The funding agencies supporting this work are (in alphabetical order of participating countries): Belgium: Research Foundation – Flanders (FWO); Czechia: Ministry of Education, Youth and Sports (MSMT); France: French National Research Agency (ANR); Germany: Federal Ministry of Education and Research (BMBF); Ireland: Health Research Board (HRB).

**Competing interests:** The authors have declared that no competing interests exist.

Conducting simulations using participants' Fitbit data allowed for optimising the triggering rules, achieving the desired median number of sedentary and walking surveys (3.9/day for both) in similar populations. Self-initiated reports of meals and drinks yielded more reports than those prompted in time-based and event-based EMA surveys, suggesting that self-initiated surveys might better reflect actual eating behaviours. This study highlights the importance of assessing feasibility and optimising EMA protocols to enhance subsequent compliance and data quality. Conducting pre-tests to refine protocols and procedures, including simulations using participants' activity data for optimal event-based triggering rules, is crucial for successful large-scale data collection in EMA studies of physical and eating behaviours.

# Introduction

## Background

Physical and eating behaviours are cornerstones of a healthy lifestyle [1, 2]. Measuring these complex behaviours comprehensively is crucial for understanding their impact on health, enabling reliable surveillance, and developing effective interventions [3–5]. Traditionally, both physical and eating behaviours have been measured using retrospective questionnaires. However, such questionnaires have significant disadvantages, including recall bias, as they rely on participants' memory and motivation, which can lead to inaccuracies [6–8]. To overcome these limitations, physical behaviour is now often measured using accelerometers, which provide objective, continuous measurements of movement [9, 10]. Despite their advantages, accelerometers have a significant drawback: their inability to provide context for the recorded activity. For example, while they can detect movement and certain types of activities, they cannot generally ascertain the activity to a great level of specificity, nor can they provide information on important contextual factors related to the measured behaviour, such as who the individual is with.

Ecological Momentary Assessment (EMA) methods, when combined with accelerometers, can add the much-needed context to physical behaviour data [11, 12]. EMA involves real-time data collection through self-reports, allowing researchers to capture participants' activities and experiences as they occur and as individuals perceive them [13]. By integrating EMA with accelerometer data, it becomes possible to understand not only the type of activity and its intensity and duration but also the situational context, such as the environment and perceived experiences associated with the activity [14]. Similarly, EMA can improve the measurement of individual determinants of eating behaviour [15] and capture real-time data on food and beverage consumption [16]. Overall, it is a useful method helping to reduce recall bias by providing more accurate and detailed information about eating patterns, such as the timing, location, and social context of meals and snacks [17, 18]. Thus, by using EMA, researchers can potentially gain a comprehensive understanding of both physical and eating behaviours and their interrelations.

Time-based EMA involves prompting participants to complete surveys at predetermined intervals, regardless of their current activity [19]. This method is particularly relevant for examining the determinants of behaviour, as it allows for systematic sampling and captures routine contexts, thus offering a comprehensive overview of the day [20]. Recent advances in sensor technologies have introduced the novel concept of event-based EMA, where surveys are dynamically triggered by specific events detected by sensors, offering a more context-aware

alternative to traditional time-based approaches [21, 22]. For instance, sensors embedded in wearable activity trackers can now detect prolonged bouts of sedentary behaviour [23] or episodes of walking and prompt participants to complete surveys based on these events [24]. This method aims to enhance the relevance and timeliness of the data collected, as it ties the self-reports directly to significant behavioural events [25]. Combining both event-based and time-based EMA can provide a more comprehensive understanding of participants' behaviours and contexts.

## Challenges in implementing EMA

While employing EMA offers numerous benefits, it also presents several challenges [26]. These include ensuring participant compliance with the protocol and managing the burden of frequent survey prompts, which can result in response fatigue and a habituation effect [21].

Suboptimal compliance and the resulting missing data can significantly compromise the validity and reliability of study findings. For example, a meta-analysis of 68 studies on various health-related behaviours and psychological constructs in adults estimated an overall compliance rate (proportion of completed surveys) of 82%, ranging from 38% to 98% [27]. The compliance rate was associated with the number of prompts per day and items per prompt: in non-clinical populations, prompting 1 to 3 surveys per day was associated with higher compliance compared with studies with more than 3 surveys per day and surveys with more than 26 items had lower compliance compared with surveys with ≤26 items [27]. Furthermore, a recent study summarising person-level data from four independent EMA studies, which used both time-based and sensor-triggered event-based prompting schedules to explore physical and eating behaviours among 278 older adults, found a compliance rate of 75%, varying from 65% to 83% across four studies, with variations among participant subgroups and at different times of the day [28]. Additionally, research in young adults on physical behaviours reported response rates ranging from 54% to 95% across studies, highlighting the variability in compliance depending on the study design and population [29]. While the reviews provide important insights about expected compliance rates and factors associated with them, the observed variations in compliance across previous studies underscore the importance of assessing feasibility and applying tailored approaches to optimise EMA protocols for the specific population before the study commences.

Another challenge specifically relates to event-based EMA, which requires fine-tuning the rules for detecting events of interest and triggering an optimal number of prompts. For example, setting the rules too stringently (e.g., defining walking episodes as at least 20 minutes of >100 steps per minute without any outliers) will reduce sensitivity and likely lead to very few triggers in a general population [30]. Conversely, rules that are too lax can trigger an excessive number of prompts. Although the total number of prompts per day can be limited, this approach reduces specificity and may result in prompts being concentrated earlier in the day. Therefore, fine-tuning these rules is important to balance sensitivity and specificity, ensuring that event-based EMA effectively captures meaningful episodes without overburdening participants. However, optimising the rules is challenging due to the high heterogeneity in physical behaviour patterns across populations. Thus, collecting physical behaviour data from a sample of the population and using it to simulate the number of triggers and optimise the rules accordingly appears to be a promising approach.

Finally, it is not clear what EMA approach is the most suitable for reporting meals, snacks, and drinks. In a review of 40 studies, 22 studies used EMA prompts to notify participants to report their food consumption, while 15 studies instructed participants to self-initiate reports of food and beverage intake during or immediately after consumption; the remaining 3 studies

combined both approaches [31]. Another review of mobile-based EMA studies in young adults also demonstrated the predominance of prompted EMA: of 39 identified studies, 27 used prompted EMA, 7 studies used self-initiated EMA, and 4 studies combined both approaches [32]. However, none of these reviews indicated whether self-initiated or prompted EMA is more effective for reporting meals, snacks, and drinks, and the only study that directly compared both approaches in a crossover design was inconclusive [33]. Consequently, it remains unclear which method leads to better reporting of consumption frequency, timing, and types of meals, snacks, and drinks.

## Objectives

The objective of this study was to evaluate the feasibility of an EMA using both time- and event-based triggers, optimise the EMA protocol, and provide recommendations for future use in large-scale EMA data collections. Specifically, we aimed to (1) assess participants' compliance with the EMA protocol, explore reasons for suboptimal compliance, and suggest solutions for improving it; (2) fine-tune the rules for triggering the event-based EMA surveys to achieve their optimal number per day; (3) compare different EMA approaches to reporting episodes of eating behaviour (meals, snacks and drinks) and choose the one that better reflects actual eating behaviour. To address these aims, we combined quantitative analysis of EMA data with qualitative insights from post-EMA feedback interviews.

## Methods

### Study design

This study took place from October 2022 to March 2023 in five study centres from four countries: Ireland (Limerick), Germany (Bremen), France (Paris), and Czechia (Prague and Ostrava). The study included a 9-day free-living EMA data collection using the HealthReact platform connected to a Fitbit tracker. The EMA protocol combined time-based (7 per day), event-based (up to 10 per day), and self-initiated surveys, each comprising 8 to 17 items. In Germany and Czechia, the data collection was followed by structured feedback interviews to gain qualitative insights into participant experiences.

The study was conducted within the WEALTH (Wearable sEnsors for the Assessment of physicaL and eaTing beHaviours) project to optimise the EMA protocol for the project purposes. The WEALTH project was initiated in 2021 with the aim of collecting EMA-labeled accelerometry data from a large sample of healthy adults across four countries. It seeks to apply machine learning methods to develop an integrated data collection system that simultaneously captures physical and eating behaviours and their interrelation. The rationale and protocol of the WEALTH project have been published elsewhere [34].

### Population

Participants were recruited from among healthy adults aged 18 to 64 through researchers' social networks, including students, colleagues, family, and friends, and from the community. We aimed to recruit at least 10 participants from each country, totalling at least 50 participants. Recruitment methods included word of mouth, flyers, emails, and social media outreach. An informational letter detailing the study's objectives, design, and purpose was provided to all potential participants. Exclusion criteria included physical impairments that could interfere with the activity protocol and employment as shift workers. Additionally, participants were required to be willing to carry and use their smartphones with a mobile data plan and engage in the entire data collection process throughout the study period. The recruitment periods

varied across study sites. In Ireland, recruitment began on March 2, 2023, and ended on March 7, 2023. In Germany, it started on October 7, 2022, and concluded on October 12, 2022. France initiated recruitment on November 24, 2022, and finished on January 18, 2023. In Czechia, recruitment commenced on October 11, 2022, and was completed by November 29, 2022. In Ireland and Czechia, participants received a monetary incentive of 20 euros upon completing the data collection.

Prior to the study's commencement, ethics committee approval was granted by the following institutions: the Education and Health Sciences Faculty Research Ethics Committee (22_09_10_EHS) in Limerick, the Ethics Committee of the University of Bremen (2022–25) in Bremen, the Comité de Protection des Personnes CPP Ile-de-France VI (2022-A02208-35) in Paris, and the Committee for Research Ethics at the University of Hradec Kralove (11/2022) in Prague and Ostrava. Participants were informed of their right to withdraw from the study at any time without providing justification. Written informed consent was obtained from all participants prior to the commencement of measurements.

## EMA platform

The EMA data were collected using the HealthReact platform developed at the University of Hradec Kralove (Czechia). HealthReact is a software suite comprising a server-side component and a mobile app for iOS and Android smartphones [35]. The HealthReact server is capable of collecting data from a diverse range of sensors, including wearable devices from Fitbit, and evaluating these data in real time to automatically trigger event-based EMA surveys based on predefined rules. These rules can be customised by researchers through a user-friendly web interface. The EMA surveys are then pushed to and displayed on the HealthReact app on participants' smartphones. Additionally, HealthReact supports traditional time-based surveys, which are triggered at random times within a specified time frame. Finally, HealthReact also enables the possibility of self-initiated surveys.

HealthReact allows the configuration of various parameters for both event-based and time-based EMA protocols. These include setting the restricted time window for prompting the survey (e.g., 8 am to 8 pm), the maximum number of surveys per day (e.g., 3 per day) or within a pre-determined time window (e.g., morning versus afternoon), the minimum interval between two surveys (e.g., 90 minutes), and the survey expiration time (e.g., 8 minutes). Specifically for event-based surveys, researchers can set the minimum duration of the event required to trigger a survey (e.g., 10 minutes), a variable of interest and its threshold (e.g., steps per minute greater than 60), time epochs for which this variable should be calculated and how (e.g., averaging across 1-minute epochs), and the number of outlier epochs or missing values that are tolerated. Furthermore, these rules can be combined so that two or more conditions must be met simultaneously (e.g., steps equal to zero and heart rate greater than 30 bpm to distinguish sedentary time from non-wear). Finally, HealthReact enables sequencing of the rules so that the fulfilment of the start rule initiates a standby mode, but the survey is only triggered after the fulfilment of the ending rule (e.g., triggering a survey after a participant stops running).

To trigger the event-based surveys within HealthReact, we chose the Fitbit Charge 5 device as a suitable wearable sensor. The Fitbit Charge 5 (Fitbit Inc., San Francisco, CA) is a versatile fitness tracker equipped with a 3-axial accelerometer for measuring movement patterns and physical activity levels. In addition to measuring physical activity, particularly steps, it also features a heart rate monitor that can be used to control for device wear. Fitbit devices have been demonstrated to accurately detect steps in minute epochs, a necessary prerequisite for triggering event-based EMA surveys [30]. Additionally, Fitbit has been shown to be reasonably accurate in detecting episodes of sedentary behaviour [30]. However, a limitation of Fitbit is that

the device automatically syncs with the Fitbit server only approximately every 15 minutes, resulting in delays in data evaluation. This delay, inherent to the Fitbit ecosystem, can be further prolonged by irregular syncs due to an unreliable internet connection or participants' smartphone usage preferences and behaviour.

To manage delayed data when triggering event-based surveys, HealthReact allows setting the time span in which events are to be sought retrospectively. For example, setting this time span to a maximum of 1 minute backwards means that the event of interest must continue until the very last minute before the sync in order to be detected. This setting is useful when aiming at answering the survey as close to the event of interest as possible. However, it also means that events occurring between syncs may be missed. Alternatively, setting the time span to a maximum of 17 minutes backwards ensures that all events are captured, assuming the regular 15-minute sync, but it also means that surveys may be prompted long (17 minutes) after the event has ended. It is worth emphasising that the issue with delayed data and unreliable internet connection only affects the event-based surveys. The time-based surveys, in contrast, are pre-scheduled by the HealthReact system for the entire study period at the start of the study, ensuring they are triggered even in the complete absence of an internet connection.

## EMA protocol

The EMA protocol was developed to align with the WEALTH project objectives by the multi-disciplinary WEALTH team, comprising experts in EMA (MJ, SE, DVD, TV), machine learning methods (CB, LS, PVV), physical activity (AD, GC, JMO), eating behaviour (AH, JH, LKF), and developers of the HealthReact platform (JK, RC). The protocol was created through an informal decision-making process that included three workshops led by TV, SE, JK, and MJ, incorporating feedback from the international WEALTH Advisory Panel (members' names listed in the Acknowledgments). During the development of the EMA protocol, the WEALTH team also determined the optimal number of event-based surveys through expert consensus to align with the project objectives of collecting a large dataset of EMA-labeled accelerometry data for the application of machine learning methods. Specifically, the optimal number of surveys was set at four per day for sedentary surveys and three per day for walking and running surveys, balancing the feasibility of collecting sufficient data with minimizing participant burden.

The protocol consisted of 9 days of free-living EMA data collection (Fig 1). The first day (day -1) served as a training day in the lab, the second day was a lead-in day (day 0) where participants familiarized themselves with the study protocol and devices in their natural environment, and the data from the subsequent 7 days were included in the analyses (days 1 to 7). Each day, participants received 7 time-based surveys and up to 10 event-based surveys. These event-based surveys were triggered by prolonged bouts of sedentary behaviour (up to 4), sustained walking (up to 3), or running (up to 3). After a survey was triggered, participants received reminders at 3, 6, and 7 minutes if the survey had not yet been filled in; the surveys expired after 8 minutes. Additionally, participants were required to report all their meals, snacks, and drinks.

The seven time-based surveys were triggered randomly within predefined time frames (morning survey: 6:00–9:45, daily surveys: 10:15–11:45, 12:15–13:45, 14:15–15:45, 16:15–17:45, 18:15–19:45, evening survey: 20:15–22:00). This configuration resulted in a minimum time interval of 30 minutes between two time-based surveys. The daily surveys included 9 items covering behaviour, affective states, fatigue and food cravings. The morning survey included the same items as the daily surveys plus 5 additional items to assess sleep duration and quality. The evening survey included the same items as the daily surveys plus 5 additional

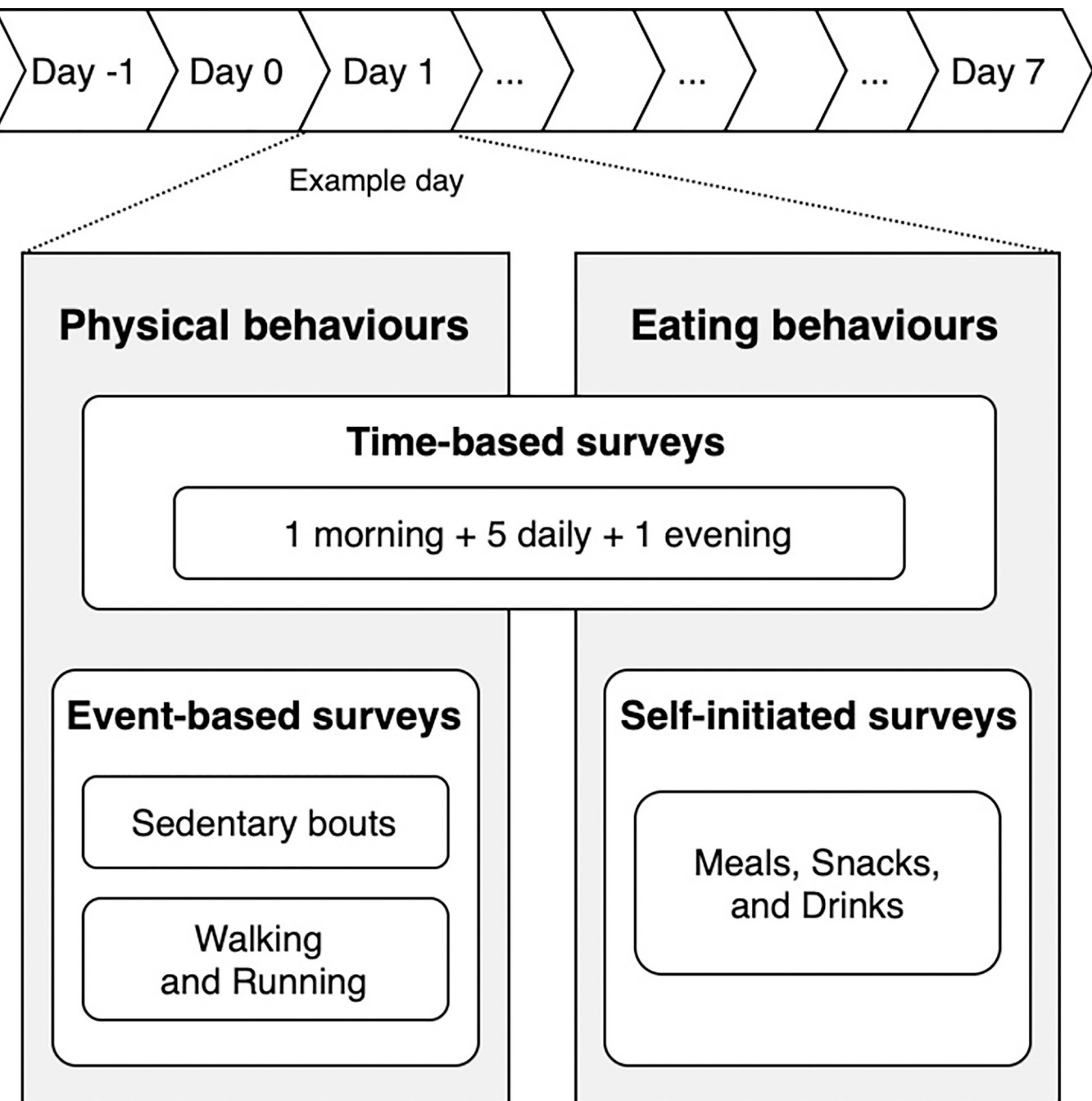

**Fig 1. Schematic overview of the EMA protocol.**

items to assess general well-being for the respective day (see S2 File for the individual survey items).

The sedentary surveys in Germany and Czechia were triggered by 30 minutes of zero steps with no outlier minute tolerated, while detecting heart rate; the time span was set at a maximum of 1 minute backwards. As the interim analysis showed that this setting resulted in a very low number of triggers, the required duration of the sedentary event was shortened from 30 to 20 minutes in Ireland and France (where the data collection was conducted later), with other

rules remaining the same. The maximum number of sedentary surveys was 2 in the time window from 8 am to 2 pm and 2 from 2 pm to 8 pm, allowing for a maximum of 4 per day. The minimum interval between two sedentary surveys was set at 90 minutes. The sedentary surveys included the same items as the time-based daily surveys plus 5 additional items covering body posture, physical behaviour, location, and company.

The walking and running surveys in Germany and Czechia were triggered by 10 minutes of 60 to 139 steps per minute [36], allowing for 2 outlier minutes above the upper threshold for walking, and 140 or more steps per minute for running; the time span was set at a maximum of 2 minutes backwards. As with the sedentary surveys, the required duration of walking and running events was shortened from 10 to 5 minutes in Ireland and France due to the low number of triggers revealed in an interim analysis. Additionally, 2 outlier minutes below the lower threshold were allowed for both walking and running surveys. The maximum number of walking and running surveys was 3 per day for each type, and the minimum interval between two surveys of each type was 90 minutes. The surveys included the same items as the daily surveys plus 8 additional items covering body posture, type and intensity of physical activity, location, and company.

The EMA reports of meals, snacks, and drinks were implemented differently in Germany and Czechia versus Ireland and France to explore which approach resulted in better reporting. In Germany and Czechia, participants were instructed to report all their meals, snacks, and drinks as soon as they finished them by self-initiating a respective survey. They were allowed to complete the surveys retrospectively but no later than midnight (self-initiated EMA). Additionally, participants were reminded in each time-based and event-based survey to report their meals, snacks, and drinks. In Ireland and France, participants were asked at the end of each time-based and event-based survey whether they had a meal, snack, or drink since the last survey. If they affirmed, they were prompted to complete the respective survey (prompted EMA). These surveys included 8 items covering the drink or snack category, location, company, and concurrent and preceding behaviour; however, participants were not asked to specify the components or foods of the meals, snacks, and drinks.

## Procedures

Eligible participants were invited to an initial lab visit (day -1), where they received the Fitbit Charge 5 tracker and were asked to use their smartphones to download the Fitbit and HealthReact apps, log into these apps using credentials provided by the researcher team, and pair the Fitbit app with the Fitbit tracker. Additionally, they were instructed to adjust their smartphone settings (mobile data always on, unmute or at least set to vibration, no battery saving mode, Fitbit notifications off, HealthReact notifications on) and to maintain these settings throughout the study. Participants who could not use their smartphone this way (e.g., did not have a suitable mobile data plan) were offered a study smartphone with an internet connection.

Once everything was set up, researchers sent a test EMA survey to ensure that each participant received a notification on their smartphone's locked screen with a ring or vibration and to allow participants to practice navigating the HealthReact app. Before leaving the lab, participants were asked to maintain their usual behaviour throughout the study, wear the Fitbit 24 hours a day (except for one night designated for recharging the device), report all their meals, snacks, and drinks, and make every effort to complete as many prompted surveys as possible. Participants were explicitly instructed to never answer surveys while driving, riding a bike, or performing activities that require full attention. On the next lead-in day (day 0), researchers checked whether participants answered the surveys. If not, they contacted them and attempted to resolve any potential issues.

After completing the subsequent seven full days of the EMA study (days 1 to 7), participants returned to the lab to return the Fitbit tracker. In Germany and Czechia, participants underwent structured feedback interviews covering topics such as the usability of their own smartphones for the study, their perceived burden of the number and length of the surveys, the occasions and reasons for missed surveys, and the challenges related to reporting meals, snacks, and drinks (see S1 File for a copy of the interview topic guide).

### Data processing and analysis

Participants' EMA data were exported from the HealthReact server as CSV files and further processed in R software (version 4.4.0). Fitbit data were exported as step count in minute epochs and individual heart rate measurements.

### Compliance outcomes

The process outcomes of the EMA surveys that reflect participants' compliance with the protocol [37] included the percentage of surveys that a participant started to respond to (response rate), time elapsed between prompting and starting to respond to surveys (latency), and time to complete surveys (time to completion). These outcomes were summarised using medians and interquartile ranges (IQRs) of participants' averages. Time to completion was only reported for Germany and Czechia, as the surveys in Ireland and France might or might not include items related to their meals, snacks, or drinks, making calculating the average completion time meaningless. The process outcomes were also analysed across individual days to assess participants' response fatigue and habituation effect. To evaluate temporal trends in these outcomes, we employed linear mixed-effects models (lme4 package in R) with the level of significance set at 0.05. Answers to the individual survey items were not analysed in this study.

### Event-based trigger optimisation

To estimate the expected number of triggers for various alternative event-based EMA settings, we conducted a series of simulations using step count and heart rate data as recorded by Fitbit (for these simulations, only data from participants who had a full 7 valid days were used, where a valid day was defined as having at least 600 minutes (10 hours) where there was at least one heart rate recording logged for each of those minutes. First, we performed a simulation using the original settings as applied in Germany and Czechia (i.e., 30 mins of zero steps, no outliers, and time span set to 1 min backwards for sedentary events; 10 mins of 60 to 139 steps, 2 outlier minutes above 139 steps allowed, and time span set to 2 mins backwards for walking events; the minimum interval between two triggers was set to 90 mins for each type of events and the triggers for sedentary events were limited to a time window from 8 am to 8 pm). It is important to note that the simulation results do not match exactly the observed number of triggers because the simulation assumes a regular 15-minute syncing pattern of Fitbit, which never happens in real life (e.g., due to a lack of internet connection). Then, we manipulated the settings (Tables 3 and 4) and re-ran the simulations to better understand how individual parameters affect the number of triggers per day. Lastly, we performed a simulation with settings that resulted in a number of triggers close to the optimal number of surveys (i.e., 4 for sedentary and 3 for walking surveys) desired for the WEALTH project.

### Meal, snack, and drink reports

The numbers of meal, snack, and drink reports were summarised using medians and interquartile ranges of participants' daily averages. The daily averages for individual participants

were calculated by summing the total number of reports over the entire study period and dividing by the denominator either 7 (representing all seven days of the study) or the number of days with at least one report of any type for the respective participants. The differences between prompted and self-initiated reports were analysed using linear models adjusted for age and sex, with the level of significance set at 0.05.

### Feedback interviews

During the structured interviews, researchers recorded the participants' answers on a pre-defined template form. Subsequently, these answers were coded, categorised and quantified as counts, providing insights into participants' experiences and potential reasons for suboptimal compliance with the EMA protocol.

## Results

### Participants

A convenience sample of 52 participants (31±9 years, 56% females) from four different countries, France (n = 11, 33±13 years, 27% females), Germany (n = 12, 35±11 years, 75% females), Ireland (n = 13, 29±7 years, 62% females), and Czechia (n = 16, 28±6 years, 56% females), participated in the study. All participants completed the study.

### Compliance outcomes

The numbers of time-based and event-based surveys, along with the respective response rates, latency, and time to completion for all 52 participants, are summarised in Table 1. Notably, due to stringent triggering rules for running events, there were zero running surveys in Czechia and only one in Germany, which was not answered.

Temporal trends in the response rate, latency, and completion time across individual days for all survey types combined are depicted in Fig 2. Over the 7 days of the study, the response rate decreased by 1.66 percentage points per day (SE = 0.46, p < 0.001), and the completion time decreased by 1.62 seconds per day (SE = 0.75, p = 0.032). There was no significant effect of the day in the study on the latency of responding (p = 0.36).

### Event-based trigger optimisation

Of the 52 study participants, 4 did not wear the Fitbit for the entire study period, and 2 additional participants did not have 7 valid days, leaving 46 participants for the simulations. Among these participants, the average daily wear time was 1,357±124 minutes, and the average daily step count was 11,189±2,881.

The original settings yielded medians (IQRs) of 2.71 (2.07–3.43) and 0.71 (0.43–1) triggers per day for sedentary and walking events, respectively. The number of triggers resulting from the manipulation of individual parameters is depicted in Tables 2 and 3 for sedentary and walking events, respectively. The settings which resulted in a number of triggers close to the optimal number of surveys desired for the WEALTH project were 20 minutes of zero steps, no outliers, and a time span set to 1 minute backwards for sedentary events; and 5 minutes of 60 to 139 steps, 2 outlier minutes below 60 steps allowed, and time span set to 17 minutes backwards for walking events. This optimal setting yielded medians (IQRs) of 3.86 (3.14–4.54) and 3.86 (3.29–4.54) triggers per day for sedentary and walking events, respectively. Simulations were not done for running events due to their near absence in the dataset.

**Table 1. Descriptive statistics of time-based and event-based EMA surveys over the 7 days of the study.**

| | All countries | Czechia | Germany | France | Ireland |
|---|---|---|---|---|---|
| **All surveys (No. per day)** | 9.8 (7.8–11.8) | 8.5 (7.7–9.2) | 7.8 (7.2–10) | 11.7 (10.7–12.3) | 11.9 (10.6–12.3) |
| Response rate (fraction) | 0.49 (0.36–0.65) | 0.6 (0.42–0.73) | 0.43 (0.26–0.55) | 0.48 (0.37–0.57) | 0.42 (0.34–0.55) |
| Latency (sec) | 127 (105–167) | 111 (76–151) | 137 (118–169) | 127 (114–152) | 166 (109–191) |
| Completion time (sec) | 67 (62–78) | 74 (66–80) | 63 (56–67) | NA | NA |
| **Time-based surveys (No. per day)** | 7 (7–7) | 7 (7–7) | 7 (7–7) | 7 (6.9–7) | 7 (7–7.1) |
| Response rate (fraction) | 0.51 (0.36–0.72) | 0.67 (0.48–0.8) | 0.44 (0.22–0.56) | 0.49 (0.33–0.67) | 0.42 (0.33–0.67) |
| Latency (sec) | 136 (84–170) | 108 (64–151) | 143 (115–169) | 148 (94–164) | 164 (73–189) |
| Completion time (sec) | 67 (62–78) | 72 (66–80) | 63 (58–67) | NA | NA |
| **Event-based surveys combined (No. per day)** | 3.1 (1.1–4.7) | 1.4 (0.9–2.2) | 1.6 (0.8–3.4) | 4.7 (3.8–5.2) | 4.9 (3.7–5.1) |
| Response rate (fraction) | 0.34 (0.17–0.6) | 0.23 (0.15–0.41) | 0.36 (0.13–0.69) | 0.45 (0.22–0.66) | 0.34 (0.22–0.69) |
| Latency (sec) | 142 (68–194) | 83 (68–142) | 92 (43–186) | 153 (107–166) | 164 (140–199) |
| Completion time (sec) | 69 (49–74) | 69 (50–78) | 70 (46–71) | NA | NA |
| **Sedentary surveys (No. per day)** | 2.4 (1–3.3) | 1.5 (0.8–2.2) | 1.6 (0.7–3.1) | 3.1 (2.3–3.3) | 2.9 (2.6–3.6) |
| Response rate (fraction) | 0.36 (0.18–0.61) | 0.3 (0.19–0.44) | 0.38 (0.13–0.68) | 0.47 (0.21–0.64) | 0.39 (0.15–0.69) |
| Latency (sec) | 128 (52–202) | 106 (44–142) | 92 (45–166) | 125 (89–203) | 171 (130–209) |
| Completion time (sec) | 69 (49–72) | 69 (50–74) | 69 (46–71) | NA | NA |
| **Walking surveys (No. per day)** | 0.9 (0.3–1.8) | 0.3 (0.1–0.3) | 0.1 (0.1–0.4) | 1.6 (0.9–2) | 1.4 (1.1–2) |
| Response rate (fraction) | 0.33 (0.04–0.63) | 0 (0–0.5) | 0.33 (0–1) | 0.38 (0.24–0.74) | 0.33 (0.1–0.5) |
| Latency (sec) | 173 (102–248) | 184 (100–252) | 332 (176–338) | 168 (105–248) | 165 (130–227) |
| Completion time (sec) | 93 (82–101) | 97 (88–101) | 90 (78–96) | NA | NA |
| **Running surveys (No. per day)** | 0.2 (0.1–0.4) | NA | 0.1 (0.1–0.1) | 0.4 (0.3–0.4) | 0.2 (0.1–0.3) |
| Response rate (fraction) | 0 (0–0.08) | NA | 1 (1–1) | 0 (0–0) | 0 (0–0.08) |
| Latency (sec) | 10 (7–14) | NA | 17 (17–17) | NA | 4 (4–4) |
| Completion time (sec) | 91 (91–91) | NA | 91 (91–91) | NA | NA |

As the surveys in Ireland and France might or might not include reports of meals, snacks, and drinks, their completion times are not reported. The table presents the medians and interquartile ranges of participants' averages.

## Meal, snack, and drink reports

A summary of the number of meal, snack, and drink reports over the entire 7-day period, separated into self-initiated (Germany and Czechia) and prompted (France and Ireland) reports for 51 participants who reported at least one meal, snack, or drink, is presented in Table 4. The numbers of self-initiated reports were significantly greater than those of prompted reports for all meals and drinks but not for snacks, regardless of whether all seven days or only the days with at least one report were considered as the denominator (Table 4). Over the 7 days of the

**Table 2. Simulations of the expected number of triggers for sedentary events.**

| Setting† | Event duration (min) | Threshold (steps) | Outliers (min) | Backwards (min) | Triggers (median (IQR)) |
|---|---|---|---|---|---|
| Original | 30 | = 0 | 0 | 1 | 2.71 (2.07–3.43) |
| Backwards 17 | 30 | = 0 | 0 | 17 | 3.21 (2.57–4) |
| Outlier 1 | 30 | = 0 | 1 | 1 | 3.29 (3–4.25) |
| Duration 20 | 20 | = 0 | 0 | 1 | 3.86 (3.14–4.54) |
| Optimal†† | 20 | = 0 | 0 | 1 | 3.86 (3.14–4.54) |

† To control for non-wear, all settings required at least one heart rate measurement per minute for the entire duration of the event. The minimum interval between two triggers was always set to 90 minutes. To avoid disturbing participants during sleep, the triggers were limited to a time window from 8 am to 8 pm.
†† The optimal setting was the same as the 'Duration 20' setting.

**Table 3. Simulations of the expected number of triggers for walking events.**

| Setting† | Event duration (min) | Threshold (steps) | Outliers (min) | Backwards (min) | Triggers (median (IQR)) |
|---|---|---|---|---|---|
| Original | 10 | 60 to 139 | 2 ($\geq$ 140 steps) | 2 | 0.71 (0.43–1) |
| Backwards 17 | 10 | 60 to 139 | 2 ($\geq$ 140 steps) | 17 | 1.14 (0.71–1.68) |
| Outliers below | 10 | 60 to 139 | 2 ($<$ 60 steps) | 2 | 1.21 (0.86–1.57) |
| Duration 5 | 5 | 60 to 139 | 2 ($\geq$ 140 steps) | 2 | 1.57 (1–2.11) |
| Optimal | 5 | 60 to 139 | 2 ($<$ 60 steps) | 17 | 3.86 (3.29–4.54) |

† The minimum interval between two triggers was always set to 90 minutes.

study, the average number of all meal, snack, and drink reports decreased by 0.17 per day (SE = 0.05, p = < 0.001), as depicted in Fig 2.

## Feedback interviews

In total, 27 participants from Czechia (n = 15) and Germany (n = 12) participated in the feedback interviews. None of the participants required the study smartphone, and all used their own devices. Of those, 14 (52%) were iPhones with the iOS operating system, and 13 (48%) were smartphones with the Android operating system. Fifteen (56%) participants reported no issues with the survey notifications on their smartphones, while 12 (44%) experienced occasional issues with notifications not appearing as expected.

Regarding the number of surveys prompted daily, 17 (63%) participants felt it was acceptable, though they often mentioned it was the maximum they were willing to answer and that it would be too much if it lasted for a longer period. Five (19%) participants found the number to be excessive, while the remaining 5 (19%) thought it was somewhat acceptable. Concerning the number of items per survey, 24 (89%) participants felt it was acceptable, 1 (4%) found it too long, and the remaining 2 (7%) thought it was somewhat acceptable. Furthermore, 14 (52%) participants were willing to complete the same number of surveys even if the number of items increased, but only 9 (33%) participants were willing to complete more surveys if the number of items was reduced.

Participants missed or were unable to complete surveys primarily on the following occasions: 15 (56%) participants at work, 9 (33%) during sports activities, 7 (26%) while driving, and 5 (19%) participants when with family or company. Less common occasions included

**Table 4. Summary of meal, snack, and drink reports.**

| | All days | | | Only days with at least one report | | |
|---|---|---|---|---|---|---|
| | Self-initiated reports | Prompted reports | p-value | Self-initiated reports | Prompted reports | p-value |
| All reports | 4.0 (2.1–5.1) | 1.4 (1–2.7) | 0.002 | 4.3 (3.3–5.3) | 2.2 (1.5–2.7) | <0.001 |
| Meals | 1.7 (1.1–2.4) | 0.7 (0.4–1.3) | <0.001 | 2.1 (1.6–2.4) | 1 (0.7–1.4) | <0.001 |
| breakfast | 0.7 (0.4–0.9) | 0.1 (0–0.3) | <0.001 | 0.8 (0.6–1) | 0.2 (0–0.4) | <0.001 |
| lunch | 0.6 (0.4–0.9) | 0.3 (0.1–0.6) | 0.003 | 0.7 (0.6–0.9) | 0.4 (0.1–0.6) | <0.001 |
| dinner | 0.4 (0.3–0.7) | 0.1 (0.1–0.4) | 0.010 | 0.6 (0.3–0.7) | 0.2 (0.2–0.5) | 0.011 |
| Snacks | 0.6 (0.1–1) | 0.4 (0.1–0.6) | 0.337 | 0.7 (0.3–1.2) | 0.5 (0.3–0.8) | 0.526 |
| Drinks | 0.8 (0.2–2.2) | 0.3 (0–0.9) | 0.024 | 1.2 (0.3–2.4) | 0.3 (0–1.1) | 0.022 |

The table summarises reports from 51 participants over a 7-day period, totalling 357 participant-days. Of these, 284 days had at least one report. In Germany and Czechia, participants were instructed to self-initiate reports of all their meals, snacks, and drinks. In Ireland and France, participants were prompted to report these items at the end of each EMA survey. The table presents the medians and interquartile ranges of participants' daily average reports.

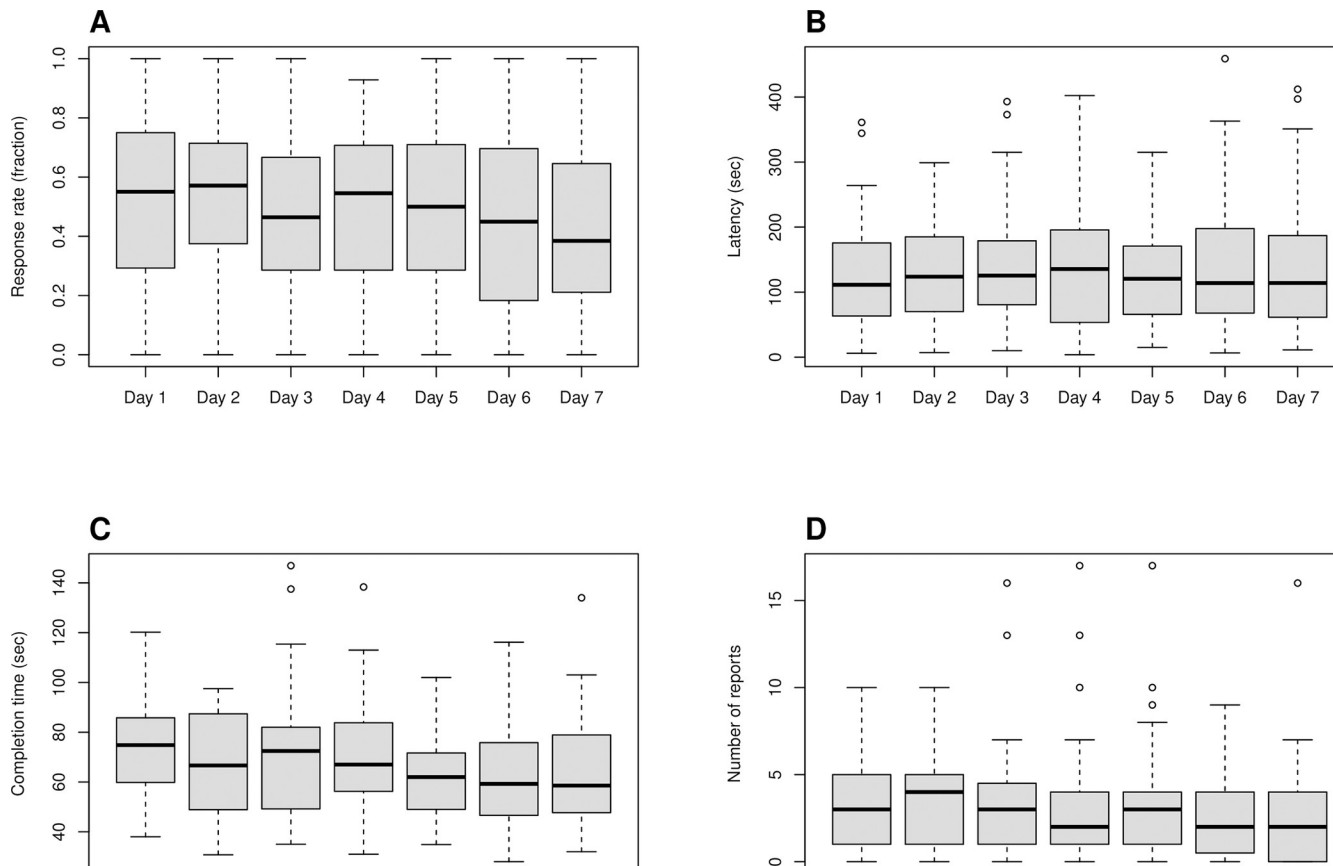

**Fig 2.** Temporal trends for response rate (A), latency (B), completion time (C), and number of meal, snack, and drink reports (D) across individual days of the study. The response rate and latency graphs combine data from all countries. The completion time graph includes data only from Germany and Czechia, as daily surveys in France and Ireland might or might not include reports of meals, snacks, and drinks. The graph of the number of meal, snack, and drink reports combines data for all types of reports from all countries.

household work, shopping, meals, and active commuting. Additionally, 2 (7%) participants received the morning surveys while still sleeping, and 2 (7%) received the evening surveys after falling asleep. Furthermore, 3 (11%) participants mentioned they would appreciate setting their own time schedule for morning and evening surveys, while 2 (7%) suggested that the surveys be triggered based on automatic sleep recognition through Fitbit. Finally, 7 (26%) participants complained that the 8-minute time window when the surveys were available for completion before they expired was too short.

Regarding meal, snack, and drink reports, it is worth noting that the feedback interviews were conducted only in the countries that used self-initiated reports (Czechia and Germany). Of the 27 participants, 13 (48%) found reporting drinks bothersome, especially when it involved just sipping water. Additionally, 2 (7%) participants mentioned that it affected their drinking habits; they refrained from drinking to avoid having to fill in a report. Furthermore, 5 (19%) participants expressed uncertainty about whether they had already reported a specific meal, snack, or drink and suggested that the HealthReact app provide an overview of reported consumption occasions. When asked about the usual time delay for reporting meals, snacks, and drinks, participants indicated that the delay varied significantly. Thirteen (48%) participants stated that they reported immediately or within one hour, while the rest experienced

delays of up to several hours, often reporting in the evening. Several participants noted that reporting drinks typically involved longer delays. Finally, 1 (4%) participant reported taking pictures to assist with reporting.

## Discussion

### Summary of the results

Our study indicated low compliance with the EMA protocol (median 49%), especially for event-based surveys (median 34%), which declined over the course of the study. This compliance rate is on the lower bound of figures reported in previous studies, where compliance rates typically exceeded 50% [27–29]. Feedback interviews revealed several issues undermining compliance, such as participants' incapability and unwillingness to answer surveys in certain contexts (e.g., at work, with family), interference with their daily schedules, too short a time to answer the surveys, and occasional technical issues resulting in missed survey notifications. These insights provide opportunities for refining the protocol to improve compliance. On the other hand, the burden of the EMA protocol, including the number of surveys and the number of items per survey, seemed to be acceptable for most participants, with some willing to accept even a slightly greater burden.

The study also showed that the rules for triggering the event-based surveys were too stringent, particularly in Germany and Czechia, resulting in far fewer surveys than desired, especially for walking surveys. The less stringent rules in Ireland and France (shortening the duration of the event from 30 to 20 minutes for sedentary events and from 10 to 5 minutes for walking events) were associated with an increased number of surveys. However, even then, the number of walking surveys was still well below the desired number, indicating the need for optimising the rules using simulations based on the existing data.

Finally, the study demonstrated that self-initiated reports of meals and drinks yield more reports than those prompted within time-based and event-based EMA surveys, indicating that self-initiated surveys might capture more occasions of eating or drinking. However, in the feedback interviews, participants criticised the need to report every sip of water, with some mentioning that it even affected their drinking habits—they preferred not to drink to avoid filling out yet another survey, indicating a potential for reactivity effects to the measurement.

### Recommendations for future EMA data collection

The results of the study were discussed at an online workshop with the multidisciplinary WEALTH team, and advice was sought from members of the WEALTH Advisory Panel. Based on these discussions, the following recommendations for future EMA data collection were made, in line with the study objectives.

**1. Improving participants' compliance.** The feedback interviews revealed that many participants were either unable or unwilling to complete surveys in various situations, such as during working hours, while participating in sports activities, and while in the company of colleagues, friends, or family members. Therefore, it was recommended that potential participants be thoroughly informed about the study requirements, and the initial training should emphasise the importance of completing as many surveys as possible. Providing suggestions on managing these situations, such as informing colleagues and family about their participation in advance, could also be beneficial. Alternatively, future studies can individualise the EMA protocol to reflect participants' constraints by tailoring the timing and frequency of surveys to accommodate each participant's unique schedule and context. This approach can significantly improve compliance, as it allows for a more flexible and participant-adapted data

collection process. However, this personalisation may lead to the underrepresentation of certain daily contexts, potentially yielding biased estimates of behaviour.

Given that a portion of participants failed to complete surveys, often due to technical issues, it was also recommended that a structured, formal, data-driven procedure for compliance monitoring should be introduced. In this procedure, participants' response rates would be monitored daily, and in cases of non-compliance, participants would be immediately contacted by a member of the research team to identify potential issues and encourage them to complete as many prompted surveys as possible. However, this approach, while beneficial for achieving higher compliance and improving data quality, can become obtrusive for participants, potentially impacting their willingness to continue with the study. Thus, each project must balance the completeness of the data against the risk of participant annoyance and early dropout if participants are contacted each time their response rate drops.

Furthermore, respecting participants' feedback that the time window when the surveys were available for completion before they expired was too short, it was recommended that the expiration be extended from 8 to 15 minutes. Further extending the expiration time could improve compliance by providing participants with greater flexibility to respond. However, this recommendation must balance providing sufficient time for participants to respond with the need to answer the survey close to the time of the trigger, which is especially important when the survey relates to specific events of interest. Similarly, introducing a snooze feature that allows participants to delay the survey could improve compliance by offering additional flexibility; however, this approach would result in responses being even further removed from the event of interest, which is why we did not recommend it.

Finally, to avoid misclassifying sleep as sedentary events and triggering the surveys inappropriately when participants were still sleeping, it was recommended that the time window for triggering sedentary episodes be shifted to start and end an hour later or, preferably, individually tailored to each participant.

**2. Fine-tuning the rules for triggering the event-based surveys.** To ensure that participants receive approximately 4 surveys on sedentary behaviour, it was recommended that the duration of the sedentary episode be shortened from 30 to 20 minutes, consistent with durations used in other studies on sensor-triggered surveys of sedentary behaviour, which have employed either 20 or 30 minutes as thresholds to balance sensitivity and specificity [24]. Other options considered included enabling one outlier minute or extending the time span to a maximum of 17 minutes backwards. However, these alternative options resulted in fewer triggers per day than deemed sufficient. Furthermore, even a short, intensive interruption of sitting (as brief as one minute) might be potentially beneficial [38]; therefore, it was preferred not to allow any outliers when triggering sedentary surveys. Moreover, extending the time span to a maximum of 17 minutes backwards could result in triggering surveys for sedentary events that actually ended more than a quarter of an hour ago, making the survey relatively unrelated to the event of interest.

To ensure that participants receive approximately 3 walking surveys, several adjustments to the original protocol had to be combined, as none of the single adjustments resulted in at least 2 surveys per day. Thus, the recommended settings for triggering walking episodes included: (a) shortening the duration of the walking event from 10 to 5 minutes; (b) allowing 2 outliers lower than the threshold of 60 to 139 steps instead of greater; and (c) extending the time span to a maximum of 17 minutes backwards. While shortening the duration of the walking events reflects that the selected sample was sedentary with rare events of uninterrupted walking for at least 10 minutes, other adjustments have implications worth considering. Allowing for outliers aligns with likely scenarios of walking in the city where individuals often stop briefly at traffic lights or shop windows. On the other hand, it is questionable whether achieving at least 60

steps in just 3 out of 5 minutes can still be considered walking. Extending the time span to a maximum of 17 minutes backwards ensures that all walking events are captured (assuming the regular Fitbit sync every 15 minutes). The downside, however, is that surveys can be triggered up to more than a quarter of an hour after the event, resulting in a somewhat loosened relation between the walking event and the survey. While our study relied on a simple threshold in steps per minute to detect walking episodes, future studies might benefit from integrating additional data sources, such as GPS, as studies combining accelerometers and GPS have shown improved accuracy and richer contextual insights into walking behaviours [39].

**3. Choosing an optimal approach to reporting meals, snacks, and drinks.**   While the design of the study did not allow for a direct comparison of self-initiated and prompted reports of meals, snacks, and drinks in a rigorous randomised setting, the substantially greater number of self-initiated reports led to the recommendation of using them as the optimal approach. However, self-initiated reports can bring their own issues, as indicated by feedback interviews where approximately half of the participants reported delays of over one hour, often reporting in the evening. Thus, the recommendation needs to be made with consideration of the study aims and population.

In addition, based on participants' feedback, it was recommended that participants not be required to report the consumption of plain water. This approach highlights the importance of simplifying reporting tasks to enhance compliance and data quality in EMA studies.

## Study limitations

There are a few limitations to our study. First, the feedback interviews were only conducted in Germany and Czechia, potentially missing important feedback from Ireland and France, especially on the prompted reports of meals, snacks, and drinks that were unique to those countries. However, since the recommendation for future data collection is to use self-initiated instead of prompted reports, this limitation is less concerning.

Second, the stringent rules set for triggering the event-based surveys resulted in a much lower number of these surveys than desired. While optimising these rules will ensure the desired higher number of event-based surveys, this increase in protocol burden can be detrimental to participants' compliance. Nonetheless, feedback from participants' interviews suggested that they are willing to accept a slightly higher burden.

Third, comparing self-initiated and prompted reports of meals, snacks, and drinks between countries instead of randomly assigning participants to either of these approaches introduces potential biases. However, given the considerable difference in the number of reports between countries with self-initiated and prompted reports, it is unlikely that the chosen approach would be inferior.

Fourth, the variability in participant compensation between countries (with incentives provided only in Ireland and Czechia) introduces a potential confound that could influence compliance rates. However, as the incentives were modest (20 euros), it is reasonable to assume they did not have a significant impact on compliance. Future studies should explore the impact of incentives in a more rigorous way, for example, by using graded compensation tied to EMA response rates, to further enhance participant compliance.

Finally, the recruitment of a convenience sample, including individuals personally familiar to the authors, may have influenced participants' motivation to complete the study and contributed to the unusually high retention rate due to a sense of interpersonal obligation.

## Conclusions

Assessing feasibility and optimising the protocol is crucial for the success of studies using EMA methods, as exemplified in the WEALTH project. Our findings underscore the

importance of addressing key issues like participant compliance, the rules for triggering event-based surveys, and the methods for reporting meals, snacks, and drinks. Without these optimisations, EMA data collection could be seriously compromised, leading to low compliance, too few event-based surveys, and suboptimal reporting of eating behaviours. By addressing these challenges through careful planning and incorporating participant feedback, researchers can markedly improve the accuracy and reliability of EMA methods, thereby enhancing their utility in studying physical and eating behaviours and their applicability in future interventions and health surveillance methods.

## Supporting information

**S1 File. Topic guide for feedback interviews.**
(PDF)

**S2 File. Individual survey items.**
(PDF)

## Acknowledgments

We are grateful for the participation of the volunteers in Czechia, France, Germany, and Ireland in this examination. We would like to express our gratitude to the WEALTH Advisory Panel for their invaluable guidance and support throughout this study. Special thanks to Genevieve Dunton, Stephanie Goldstein, Mark Hoogendoorn, Thomas Kubiak, and Stewart Trost for their expert advice and contributions. Additionally, we extend our sincere thanks to Ella Tyrrell for her exceptional project management, unwavering support, and overall contributions to the success of this study.

The WEALTH consortium members and affiliations: Alan Donnelly, Catherine Woods, Luis Sigcha, Gráinne Hayes, Pepijn Van de Ven, Daniels Stahovskis (University of Limerick, Ireland), Janas Harrington (School of Public Health, University College Cork, Ireland), Antje Hebestreit, Christoph Buck, Maike Wolters, Annika Swenne, Chandra Gowsiga Loganathan (Leibniz Institute for Prevention Research and Epidemiology—BIPS, Bremen, Germany), Jean-Michel Oppert, Leopold K. Fezeu, Jérôme Bouchan, Fabienne Delestre, Junko Kose (Sorbonne Paris Nord University, France), Hélène Charreire (Inrae, France), Greet Cardon (Department of Movement and Sports Sciences, Ghent University, Belgium), Tomas Vetrovsky, Richard Cimler, Jitka Kuhnova, Alena Faltysova (Faculty of Science, University of Hradec Kralove, Czechia), Steriani Elavsky, Veronika Horká, Michal Sebera (Department of Human Movement Studies, University of Ostrava, Czechia), Michael Janek, Dan Omcirk (Faculty of Physical Education and Sport, Charles University, Prague, Czechia).

## Author Contributions

**Conceptualization:** Michael Janek, Jitka Kuhnova, Greet Cardon, Delfien Van Dyck, Jean-Michel Oppert, Christoph Buck, Antje Hebestreit, Janas Harrington, Pepijn Van de Ven, Alan Donnelly, Tomas Vetrovsky.

**Data curation:** Michael Janek, Jitka Kuhnova, Christoph Buck.

**Formal analysis:** Michael Janek, Jitka Kuhnova, Tomas Vetrovsky.

**Funding acquisition:** Greet Cardon, Jean-Michel Oppert, Antje Hebestreit, Janas Harrington, Alan Donnelly, Tomas Vetrovsky.

**Investigation:** Michael Janek, Jitka Kuhnova, Steriani Elavsky, Leopold K. Fezeu, Christoph Buck, Luis Sigcha.

**Methodology:** Michael Janek, Jitka Kuhnova, Steriani Elavsky, Tomas Vetrovsky.

**Project administration:** Michael Janek, Jitka Kuhnova.

**Software:** Jitka Kuhnova, Richard Cimler.

**Supervision:** Richard Cimler, Steriani Elavsky, Jean-Michel Oppert, Antje Hebestreit, Alan Donnelly, Tomas Vetrovsky.

**Visualization:** Michael Janek, Jitka Kuhnova, Tomas Vetrovsky.

**Writing – original draft:** Michael Janek.

**Writing – review & editing:** Michael Janek, Jitka Kuhnova, Greet Cardon, Delfien Van Dyck, Richard Cimler, Steriani Elavsky, Leopold K. Fezeu, Jean-Michel Oppert, Christoph Buck, Antje Hebestreit, Janas Harrington, Luis Sigcha, Pepijn Van de Ven, Alan Donnelly, Tomas Vetrovsky.

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
