## [Decision Letter · Decision Letter 0]

17 Dec 2024

PONE-D-24-30072Ecological momentary assessment of physical and eating behaviours: the WEALTH feasibility and optimisation study with recommendations for large-scale data collectionPLOS ONE

Dear Dr. Vetrovsky,

I have reviewed your manuscript "Ecological momentary assessment of physical and eating behaviours: the WEALTH feasibility and optimisation study with recommendations for large-scale data collection" and the comments from three reviewers. The reviewers found your manuscript to be thorough, well-written, and valuable for the field. They particularly appreciated your novel approach using Fitbit for event-triggered EMA and your clear recommendations for future research.

The reviewers have suggested some minor revisions that would strengthen the manuscript.

Given that these revisions are relatively straightforward and do not require major restructuring, I recommend a minor revision. Please ensure that your decision is justified on PLOS ONE’s publication criteria and not, for example, on novelty or perceived impact.

We look forward to receiving your revised manuscript.

Kind regards,

Yury Shevchenko

Academic Editor

PLOS ONE

2. Thank you for stating the following financial disclosure:  [The WEALTH Project is funded by the Joint Programming Initiative (JPI) Healthy Diet for a Healthy Life (HDHL), a research and innovation initiative of EU member states and associated countries, under grant agreement No 727565, under STAMIFY (Standardised measurement, monitoring and/or biomarkers to study food intake, physical activity and health). The funding agencies supporting this work are (in alphabetical order of participating countries): Belgium: Research Foundation – Flanders (FWO); Czechia: Ministry of Education, Youth and Sports (MSMT); France: French National Research Agency (ANR); Germany: Federal Ministry of Education and Research (BMBF); Ireland: Health Research Board (HRB).  ].  Please state what role the funders took in the study.  If the funders had no role, please state: "The funders had no role in study design, data collection and analysis, decision to publish, or preparation of the manuscript." If this statement is not correct you must amend it as needed. Please include this amended Role of Funder statement in your cover letter; we will change the online submission form on your behalf.

3. Thank you for uploading your study's underlying data set. Unfortunately, the repository you have noted in your Data Availability statement does not qualify as an acceptable data repository according to PLOS's standards.

Reviewers' comments:

Reviewer's Responses to Questions

**Comments to the Author**

1. Is the manuscript technically sound, and do the data support the conclusions?

Reviewer #1: Yes

Reviewer #2: Yes

Reviewer #3: Yes

2. Has the statistical analysis been performed appropriately and rigorously? 

Reviewer #1: Yes

Reviewer #2: Yes

Reviewer #3: Yes

3. Have the authors made all data underlying the findings in their manuscript fully available?

Reviewer #1: Yes

Reviewer #2: No

Reviewer #3: Yes

4. Is the manuscript presented in an intelligible fashion and written in standard English?

Reviewer #1: Yes

Reviewer #2: Yes

Reviewer #3: Yes

5. Review Comments to the Author

Reviewer #1: The submitted manuscript is a thorough investigation of the feasibility of an intensive EMA protocol measuring eating, drinking, walking, and sedentary bouts via both self-report surveys and wearable device. The authors use a combination of Fitbit and self-report surveys to identify physical activity objectively and prompt users to report on their physical activity in event-based surveys. The manuscript is refreshingly neutral and straightforward about the authors’ findings on feasibility, with helpful details on methodology and constructive suggestions for replicating these methods and applying these findings to future investigations. The methods employed are varied, sometimes without clear rationale, and more clarity is needed on what exactly was measured within the EMA surveys.

Abstract:

It would aid reader understanding to specify what was being measured in the EMA protocol (that is, what physical and eating behaviors?)

Introduction:

It is novel that the authors use Fitbit to identify events when an event-contingent survey is indicated. In many past studies, event-contingent EMAs are essentially self-initiated; a participant is told to initiate a survey when they do a particular activity, but there is usually no means to verify that they actually did. The authors’ use of Fitbit to address this weakness in EMA methodology is novel and worth additional emphasis.

When the authors review compliance rates found in prior literature, it should be specified if these are time-based or event-based assessment compliance rates. They were likely time-based, since the authors’ use of Fitbit to identify events and associated surveys is novel.

Method:

The days are sometimes called Day 1, Day 2, etc. and other times Day -1, Day 0, etc.

We need more information about the content assessed within the surveys. What were the questions asked? Some more information is needed for readers to understand how similar their own protocols are to the one tested in this study.

Why did the activity (sedentary, walking, and running) surveys need to be done so closely to the actual event (1 or 2 minutes after, with only 8 minutes allowed before the survey expired)? Could the items not be assessed retrospectively (e.g., “when you were inactive recently…”)? Perhaps I would understand better if I knew exactly what was being measured in the surveys.

Results:

Am I understanding correctly that the authors aren’t reporting completion times for France and Ireland because the times would be variable depending on how many questions the participant answered affirmatively about meals/drinks? Given that many other researchers may include assessment of meals/drinks in their surveys in a similar way, I think it could still be valuable to report completion times if you have the data.

Why was the optimal number of surveys 4/day for sedentary events and 3/day for walking events? With the 90 minute interval between them, it seems it would be difficult for participants to have that many in a day. To assess sedentary behavior 4x/day, the authors define a sedentary episode as no movement for only 20 minutes, which for many people with desk jobs is not very long. I can imagine participants being irritated by the implication that they are not moving enough after just 20 minutes of inactivity.

Discussion

In the section “improving participants’ compliance”, it is not mentioned that lengthening the time limit allowed for survey completion would also improve compliance.

There is a missing reference at line 605.

Line 637 Another way of addressing the participants’ reasonable concern that reporting sips of water was overly burdensome is to ask participants to report when they start and finish a drink, and assume that they were sipping it in between.

Limitations

I do think a noteworthy limitation of the study is the recruitment of a convenience sample of people personally familiar to the authors. The sample’s motivation to complete the surveys and the study in general could be different than a general research participant sample. For example, all participants completed the study, which is unusual and may be due to an interpersonal sense of obligation.

I would also add that not fully measuring what was consumed in meals/drinks (if I understood the method correctly) makes this study a bit less helpful in understanding how burdensome and feasible it would be to actually measure food/drink consumed with an EMA protocol.

Reviewer #2: This study evaluates a pilot or feasibility study to optimize the study design for a large-scale EMA-study conducted in different European countries (WEALTH Project). It focused on the compliance rate and on the design of event- and time-based EMA assessments (sampling scheme). The paper deepens our understanding of some challenging issues (e.g. compliance with EMA protocol, managing the burden of frequent prompts) that are highly relevant to obtaining a reliable and credible dataset when conducting an EMA-study.

The paper is clear and provides useful information for the readers. It makes important points that are widely applicable. I just have a few issues – mainly for the discussion section – that need to be addressed before it’s ready for publication.

Introduction

I would recommend adding some information about triggered EMA designs, since you are referring to these particular types of event-based EMA assessments. Furthermore, there are already studies on triggered EMA designs that have addressed specificity and sensitivity, which could strengthen your argumentation that we need feasibility studies to “fine-tuning” the EMA-protocol.

Method

Provide some information about how many people you will need for your feasibility study and a rationale for how the number of participants will be distributed among the four different countries. The EMA-protocol varies by country, and this may be one reason for how many participants you recruit in each country. But that should be explained more clearly.

Results

Please provide detailed information on the descriptives of the samples you combined or analyzed separately. For example, if feedback interviews were only conducted in Czechia and Germany, what was the gender distribution in these subsamples? Also, do you have information on BMI or the general volume of physical activity to further describe your sample in terms of key behaviors.

Discussion

There are already some feasibility studies of EMA-protocols addressing compliance rate and burden of participants in EMA-studies that should be mentioned to integrate your findings into this area of research.

There are also some studies on triggered e-diary assessments of sedentary behavior (e.g. JMIR mHealth and uHealth - Accuracy of Sedentary Behavior–Triggered Ecological Momentary Assessment for Collecting Contextual Information: Development and Feasibility Study) or walking behavior (e.g. JMIR Formative Research - Analyzing Person-Place Interactions During Walking Episodes: Innovative Ambulatory Assessment Approach of Walking-Triggered e-Diaries) that you could use to categorize and evaluate your findings (in section 2. Fine-tuning the rules for triggering the event-based surveys)

You mentioned difficulties in assessing different physical activity behaviors (e.g. sedentary behavior, sleep). Although it is beyond the scope of your paper to provide a detailed validation study, you should discuss the advantages and disadvantages of using the Fitbit to achieve your goals of assessing different types of physical activity behaviors.

Some participants received compensation, and some did not. Based on the results, compensation does not appear to have a significant effect on compliance. But you didn’t address this issue in your results. Because compensation is an important issue in conducting intensive longitudinal studies, I recommend addressing this issue in the Method section (e.g. provide some analysis of whether compensation influences compliance) and discussing it in the Discussion section. You also address this issue in your feedback interview. Perhaps you could add some qualitative statements about how participants feel about appropriate compensation.

Reviewer #3: This was a very helpful and rigorous paper outlining challenges associated with EMA sampling schemes for health behaviors in free-living populations. The recommendations at the end are well-supported by the researchers’ findings and are an important contribution that will help to inform future research. I've included minor feedback/edits below:

Line 164: it is unclear what “target number” is referring to. Do the authors mean “maximal compliance” (i.e., 100% of triggered EMA are answered), or something else?

Lines 192-196: why were participants not screened for habitual activity levels to ensure that at least some vigorous activity occurred, if they were asked to perform their usual behavior? Also, did they screen for mobile data plan?

Line 251-252: The authors mention that lack of internet connectivity can interfere with Fitbit syncing. Were data not uploaded using cellular data—WiFi only? And could event-contingent EMA only be received when WiFi present, or could they also be received via cellular data?

Also-- Were participants asked to wear Fitbit overnight? And for how long during the day? Can the authors provide data on average wear time, step count, SB volume, and walking volume from the Fitbit?

Line 277: What do the researchers mean by a “lead-in day”?

Line 381: “valid day required at least 10 hours with at least one heart rate recording”—do the authors mean to say “one heart rate recording per minute”? Otherwise, why is the criteria for wear time so low wrt heart rate; and 10 h of what type of data (e.g., nonzero triaxial readings)?

A table with participant characteristics at baseline would be helpful.

Lines 417-418: Could this also be due to a lack of running activity in the sample (i.e., sample is fairly inactive), rather than the triggering rules?

Lines 445-446: How did the authors decide on the “optimal number of surveys” for the project? It’s a little unclear how these target numbers were chosen. Also, did the authors determine the degree of sensitivity & specificity associated with the optimal settings (I assume EMA response data captured self-reported activity that could be used to assess this)?

Page 30, Improving participants’ compliance section: Other possible recommendations could include a “snooze” feature (e.g., option to silence and delay the prompt for another 30 min, like an alarm clock), or possibly just more monetary incentive tied to EMA response rates (e.g., graded compensation based on percentage of EMA completed).

Line 605: reference needs to be fixed.

Limitations section should include the use of convenience sampling from existing social networks (might lead to greater social desirability bias during exit interviews). Also, the study provided only a small compensation (20 euros) for some participants (Ireland and Czechia), whereas others had no monetary incentive, and this might have contributed to the low compliance rate.

6. PLOS authors have the option to publish the peer review history of their article (what does this mean?). If published, this will include your full peer review and any attached files.

Reviewer #1: **Yes: **Brittany Stevenson, PhD

Reviewer #2: No

Reviewer #3: **Yes: **Rachel Crosley-Lyons, MS

---

## [Author Response · Author response to Decision Letter 0]

26 Dec 2024

Please see the attached Response letter.

---

## [Decision Letter · Decision Letter 1]

22 Jan 2025

Ecological momentary assessment of physical and eating behaviours: the WEALTH feasibility and optimisation study with recommendations for large-scale data collection

PONE-D-24-30072R1

Dear Dr. Vetrovsky,

We’re pleased to inform you that your manuscript has been judged scientifically suitable for publication and will be formally accepted for publication once it meets all outstanding technical requirements.

Your paper makes an important contribution to the field by providing practical guidance for implementing EMA protocols in behavioral research. The comprehensive evaluation of compliance issues, trigger optimization, and eating behavior assessment approaches will be valuable for researchers planning similar studies.

Kind regards,

Yury Shevchenko

Academic Editor

PLOS ONE

Reviewers' comments:

Reviewer's Responses to Questions

**Comments to the Author**

1. If the authors have adequately addressed your comments raised in a previous round of review and you feel that this manuscript is now acceptable for publication, you may indicate that here to bypass the “Comments to the Author” section, enter your conflict of interest statement in the “Confidential to Editor” section, and submit your "Accept" recommendation.

Reviewer #2: All comments have been addressed

Reviewer #3: All comments have been addressed

2. Is the manuscript technically sound, and do the data support the conclusions?

Reviewer #2: Yes

Reviewer #3: Yes

3. Has the statistical analysis been performed appropriately and rigorously? 

Reviewer #2: Yes

Reviewer #3: Yes

4. Have the authors made all data underlying the findings in their manuscript fully available?

Reviewer #2: Yes

Reviewer #3: Yes

5. Is the manuscript presented in an intelligible fashion and written in standard English?

Reviewer #2: Yes

Reviewer #3: Yes

6. Review Comments to the Author

Reviewer #2: Well done!

The authors have adequately addressed my comments

I haven't go any other suggestions or recommendations

Reviewer #3: (No Response)

7. PLOS authors have the option to publish the peer review history of their article (what does this mean?). If published, this will include your full peer review and any attached files.

Reviewer #2: **Yes: **Prof. Dr. Martina Kanning

Reviewer #3: **Yes: **Rachel Crosley-Lyons

---

## [Editor Report · Acceptance letter]

31 Jan 2025

PONE-D-24-30072R1 

PLOS ONE

Dear Dr. Vetrovsky, 

I'm pleased to inform you that your manuscript has been deemed suitable for publication in PLOS ONE. Congratulations! Your manuscript is now being handed over to our production team.

Kind regards, 

on behalf of

Dr. Yury Shevchenko 

Academic Editor

PLOS ONE